# Determinants of the intensity of uremic pruritus in patients receiving maintenance hemodialysis: A cross-sectional study

Jian-Hui Zhao[1,2]*, Qiu-Shuang Zhu[1,2], Yi-Wen Li[3], Li-Li Wang[2]

1 Department of Nephrology, Xinchang County People's Hospital, Xinchang, Shaoxing, Zhejiang Province, China, 2 Blood Purification Center, Xinchang County People's Hospital, Xinchang, Shaoxing, Zhejiang Province, China, 3 Department of Nephrology, Zhejiang People's Hospital, Hangzhou, Zhejiang Province, China

* lyzhaojh@126.com

**Data Availability Statement:** All relevant data are within the paper and its Supporting information files.

## Abstract

### Background

Uremic pruritus (UP) is a common and frustrating symptom in patients receiving hemodialysis (HD). The majority of patients have mild to moderate itching of the skin, and a small percentage have severe itching, which seriously affects their quality of life and survival rate. However, little is known about factors that influence the intensity of itching in patients.

### Methods

A cross-sectional study on uremic pruritus in male and female patients receiving HD was conducted in September 2019. This study included 148 eligible patients who received HD at the Blood Purification Center of Xinchang County People's Hospital, Zhejiang Province, China from March 2019 to June 2019. We collected general data consisted of age, sex, body mass index (BMI), place of residence, educational level, diabetes mellitus status and duration of HD; as well as clinical, biochemical indicators, including serum calcium (Ca), serum phosphorus (P), serum albumin (ALB), haemoglobin (Hb), serum intact parathyroid hormone (iPTH), pre-dialysis serum urea nitrogen (BUN), normalized protein catabolic rate (nPCR), urea nitrogen clearance index (KT/V), ferritin (FER) and pre-dialysis serum creatinine (sCR). We also assayed the inflammatory cytokine serum high sensitivity C-reactive protein (hs-CRP). The Five-Dimensional Itching Scale (5DIS) was used to evaluate the degree of skin itching (none, mild, moderate, or severe). We used multiple logistic regression to analyze influencing factors on the degree of skin itching in patients with UP.

### Results

Of the 148 patients, 60 had uremic pruritus (incidence rate, 40.54%). These included 22 cases of mild skin itching (14.86%), 30 of moderate skin itching (20.27%), and 8 of severe skin itching (5.41%). Compared with uremia patients without skin pruritus, patients with UP had higher levels of iPTH, Hb, BUN, nPCR, and hs-CRP. The composition ratio showed significant differences between urban and rural patients with different degrees of skin itching (P

**Funding:** The authors received no specific funding for this work.

**Competing interests:** The authors have declared that no competing interests exist.

= 0.017); moreover, the difference of iPTH and hs-CRP levels were statistically significant ($P$ = 0.009 and < 0.001, respectively). Using no itching as a reference, multiple logistic regression analysis showed that as hs-CRP level increased, the patient's risks of mild skin itching (odds ratio [OR] = 1.740; 95% confidence interval [CI], 1.061–2.854; $P$ = 0.028), moderate skin itching (OR = 2.8838 95% CI, 1.744–4.718; $P$ < 0.001), and severe skin itching (OR = 9.440; 95% CI, 3.547–25.124; P < 0.001) all increased as well. Compared with urban residents, rural residents have a higher risk of moderate itching (OR = 3.869; 95% CI, 1.099–13.622; P = 0.035).

## Conclusion

Levels of hs-CRP were associated with the intensity of skin itching in patients with UP. Higher hs-CRP levels were closely related to severe skin itching. The relationship between the intensity of skin itching and the environment in maintenance hemodialysis patients needs further clarification.

## Introduction

### Background

Itchy skin is a common and distressing symptom in patients with chronic kidney disease. Uremic pruritus mainly manifests as skin itching of varying degrees, either systemically or locally, of which the back, extremities, chest, and head are common. Itching appears as paroxysmal episodes of varying duration. Those with milder symptoms can have intermittent attacks, lasting several minutes each time, while those with more severe symptoms last longer, and the symptoms are usually most apparent at night [1]. Epidemiological data indicated that approximately 40% of patients with end-stage renal disease (ESRD) experience moderate to severe itching [2]. The Dialysis Outcomes Practice Patterns Study (DOPPS), conducted from 2012 to 2015, showed that 26%–48% of patients had at least moderate skin itching, whereas 13%–26% of patients had severe or extreme itching [3]. The higher the degree of skin itching, the higher the mortality rate [4]. Several causes of or factors contributing to uremic pruritus (UP) have been proposed, including increasing systemic inflammation; abnormal levels of serum parathyroid hormone (PTH), serum calcium (Ca), and serum phosphorus (P); opioid receptor imbalance; and neuropathological processes [2]. However, the factors affecting the intensity of skin itching in patients with uremia are still unclear. We used a five-dimensional itching scale [5] to evaluate skin itching across multiple dimensions in uremic patients undergoing maintenance HD. We also included general patient data consisted of age, sex, body mass index (BMI), place of residence, educational level, diabetes mellitus status, and duration of HD; as well as clinical indicators comprised of Ca, P, serum albumin(ALB), Hemoglobin(Hb), PTH, serum creatinine(sCR), serum urea nitrogen (BUN), urea nitrogen clearance index (KT/V), ferritin (FER) before dialysis, and hs-CRP to analyze the determinants of Itching in these patients.

### Objectives

1. Explore the prevalence of UP in maintenance HD patients and the demographic and clinical characteristics of pruritus patients.

2. Analyze the relationship between high-sensitivity C-reactive protein and other factors and the intensity of skin itching.

## Methods

### Study design

This study is a cross-sectional study on UP in patients receiving HD. The Ethics Committee approved this study of Xinchang County People's Hospital, Zhejiang Province, China (Approval No. XCXRMYY2018-001), and all patients signed written informed consent.

### Study setting

We included a total of 148 subjects who received HD at the Blood Purification Center of Xinchang County People's Hospital, Zhejiang Province, China from March 2019 to June 2019. each of them signed an informed consent form. Collection of data was by the authors or research personnel, and all personnel received training and guidance from the corresponding author. Data statistics and analysis were from September to December 2019.

### Participants

Inclusion criteria were as follows: (1) ≥1 month of HD in dialysis units and (2) age 20–90 years. Exclusion criteria were as follows: (1) primary skin diseases (e.g., eczema, psoriasis, neurodermatitis, allergic dermatitis, and drug rash), (2) use of right-cyclic saccharin, analgesics, antibiotics, or lipid-lowering drugs, (3) peripheral neuropathy, thyroid disease, leukaemia, lymphoma, bile siltation, liver lesions, or pregnancy, (4) patients with hearing/communication impairments who could not complete the study, and (5) patients with infections, malignant tumours, cardiovascular disease, or tissue damage. These criteria were applied to each patient.

### Variables

The diagnostic criteria for uremic pruritus adopted internationally are as follows: (1) uremia patients exclude skin pruritus caused by other diseases; (2) itching occurs at least three days within two weeks, and itches several times a day, each time itching It lasts a few minutes and affects the patient's life; (3) Itching in a specific pattern lasts for more than six months [6]. We used the Five-Dimensional Itching Scale (5DIS) to assess itching based on the five dimensions of degree, duration, direction, disability, and distribution. The total score ranged from 5 (no itching) to 25 (most severe Itching). Patients with scores <5 points were defined as having no itching; those with scores 6–25 points were considered itch patients. The score of 6–10 points indicated mild itching, 11–20 points marked moderate itching, and 21–25 points indicated severe Itching. Diagnosis of uremia pruritus meets the following conditions: 1. The total score of pruritus, according to 5DIS is between 6–25 points. 2. Meet the international diagnostic criteria for uremic pruritus (2) and (3). 3. Meet the exclusion criteria. ultimately, we included a total of 60 patients with UP and 88 patients without UP. Body Mass Index (BMI) was calculated using the following formula: weight (kg) / height (m)$^2$. Lean patients were defined as those with BMI ≦18.4 kg/m$^2$, patients with normal BMI 18.5–23.9 kg/m$^2$, heavy patients with BMI 24.0–27.9 kg/m$^2$, and obese patients with BMI ≧28.0 kg/m$^2$. According to the World Health Organization (WHO) standards, elderly patients were defined as those age ≧65 years, and non-elderly patients as those age <65 years.

### Data source/Measurement

Patients data included of Sex, Age, BMI, place of residence (rural or urban), educational level (illiterate or semi-literate, primary, junior high school, high school, or university or above), diabetes mellitus status (present or absent) and duration of maintenance HD (Months) were obtained by the investigator interviewing the patients. We assayed biochemical indicators

comprised of Ca, P, iPTH, Hb, ALB, sCR, Bun, nPCR, FER, KT/V and hs-CRP. The hospital's central laboratory performed all of the laboratory tests, and auto-analyzers were used to determine biochemical data, iPTH was measured with Roche second-generation assay. The clinical biochemical indicators were based on the data within three months from the survey day. If there is no data within three months, the patient is required to re-test. If the patient is unwilling to cooperate with the test, it was considered a missing value. $spKT/V$ ($single\ pool\ KT/V$) = $-ln(R - 0.008t) + (4 - 3.5R) * (\Delta BW/BW)$. Note: K refers to Blood urea clearance rate of dialyzer (L/h), T is the dialysis time (h), and V is the distribution volume of urea (V). R is the ratio of blood urea nitrogen after dialysis to blood urea nitrogen before dialysis; t is single dialysis time in h; $\Delta BW$ is the weight change value before and after dialysis, ultrafiltration, unit L; BW is weight in kg. Blood collection requirements: blood samples before dialysis from the artery end of the vascular path, after dialysis before blood sample collection to stop ultrafiltration, reduce blood flow of 50 ml/min, wait 15 seconds after blood collection from the artery as a blood sample after dialysis. $nPCR(g/kg/d) = C_0/[a + b \times KT/V + c \div (KT/V)] + 0.168$ [7], where $C_0$ indicates pre-dialysis blood urea nitrogen in mg/dl (1mmol/L urea nitrogen is equal to 2.802mg/dl). a, b, c has different coefficients depending on the time of the dialysis schedule. In our blood purification center, patients are dialyzed three times a week, either on Mondays, Wednesdays, Fridays or either Tuesdays, Thursdays, Saturdays, so that the time between the first dialysis sessions at the beginning of the week is longer. The following formula is used: beginning-of-week: $nPCR(g/kg/d) = C_0/[36.3 + 5.48 \times KT/V + 53.5 \div (KT/V)] + 0.168$. https://dx.doi.org/10.17504/protocols.io.bp94mr8w [PROTOCOL DOI].

## Sample size

Epidemiological data indicated that approximately 40% of patients with end-stage renal disease (ESRD) experience UP [2]. The overall prevalence of uremia and pruritus is 0.4, and the tolerance is set to 0.1, taking a = 0.05, the total amount of hemodialysis in our blood purification center is 180 patients. According to PASS 15 Power Analysis and S ample Size Software (NCSS, LLC. Kaysville, Utah, USA, ncss.com/software/pass), the calculated sample size is 121 people, assuming that the dropout rate is 20%, and the total sample size is 152 people.

## Statistical methods

We performed statistical processing using line data from SPSS software version 24.0 (IBM Corp., Armonk, New York, US). Quantitative data from a normal distribution are expressed as mean ± standard deviation and comparison between groups were made using one-way analysis of variance or t-test. Quantitative data from a non-normal distribution are expressed as medians and interquartile ranges and comparison between groups of continuous variables were made using a non-parametric test. comparison between groups of classification variables was based on the rank-sum or chi-square test. A multiple logistic regression model analyzed the factors associated with the degree of skin itching in patients with uremia. All variables were diagnosed by collinearity. After elimination of the collinearity problem, the variables with statistical significance in the univariate analysis and the variables with statistical differences between the groups of different degrees of skin pruritus were included in the multivariate analysis (P<0.05 considered statistically different). Finally, we had four factors, including Hb, PTH, hs-CRP, place of residence in the multiple logistic regression model. P < 0.05 was statistically significant. In this study, the variables iPTH, hs-CRP, BUN, KT/V have two missing values, respectively, and therefore a total of 8 missing values. The analysis of missing values indicates that these variables were not MCAR (missing completely at random), so this study uses the EM (expectation-maximization) algorithm to fill in.

## Results

### Participants

Our blood Purification Center has a total of 180 patients. 20 patients were excluded, and 10 of them were unwilling to participate in the survey. 5 were peritoneal dialysis combined with hemodialysis, but hemodialysis was only treated once a week, and the remaining 5 were scheduled for 2 dialysis a week. Therefore, a total of 160 patients who regularly underwent hemodialysis three times a week were included in this study, 3 of them did not respond. Besides, we excluded 1 case of cirrhosis, 1 of dialysis allergy, 1 of drug rash, 3 of nodule itch, and 3 of hearing impairment that rendered the patient unable to communicate; ultimately, we included a total of 148 eligible subjects, in which 89 males (60.1%) and 59 females (39.9%). The median duration of HD was 42 months. There were 60 cases of UP (prevalence rate, 40.54%), including 22 mild cases (14.86%), 30 moderate cases (20.27%), and 8 severe cases (5.41%), As shown in Fig 1.

### Descriptive data

The primary clinical characteristics of the included patients are shown in Table 1.

### Outcome data

Compared with uremia patients without skin pruritus, patients with UP had higher levels of IPTH, Hb, BUN, nPCR and hs-CRP. No statistically significant differences were found in demographic characteristics such as Sex, Age, Educational level, BMI, Diabetes mellitus status or Place of residence; in levels of Ca, P, ALB, FER, KT/V, or sCR; or in HD duration (Table 2).

The composition ratio showed a difference between rural and urban patients with varying degrees of skin itching ($P = 0.017$). Differences in IPTH and hs-CRP levels of patients with different degrees of skin itching were statistically significant ($P = 0.009$ and $<0.001$, respectively), as shown in Table 3.

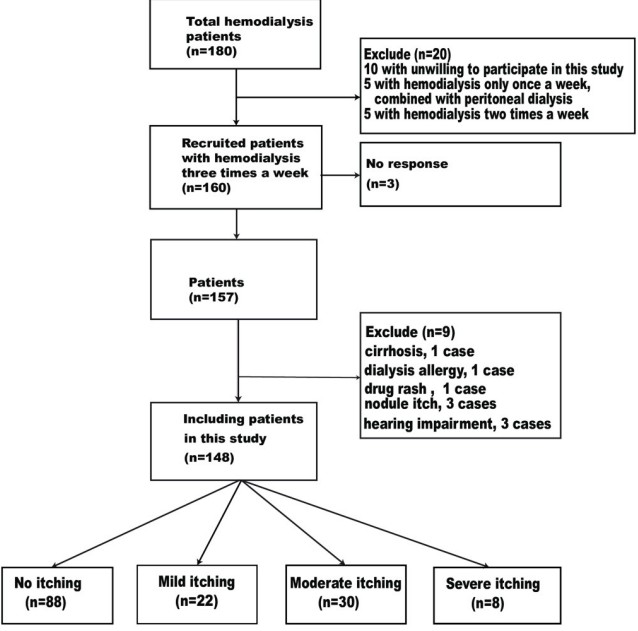

**Fig 1. Patient screening and grouping.**

**Table 1. Demographic and clinical characteristics of patients enrolled at baseline.**

| Variable | Results |
|---|---|
| Sex, n (%) | |
| Male | 89 (60.1%) |
| Age, n (%) | |
| Non-elderly | 99 (66.9%) |
| Educational level, n (%) | |
| Illiterate or semi-literate | 23 (15.5%) |
| Primary school | 36 (24.3%) |
| Junior high school | 54 (36.3%) |
| High school | 19 (12.8%) |
| University and above | 16 (10.8%) |
| Place of residence, n (%) | |
| Rural | 100 (67.6%) |
| BMI, n (%) | |
| $\leqq$18.4 | 25 (16.9%) |
| 18.5–23.9 | 101 (68.3%) |
| 24–27.9 | 15 (10.1%) |
| $\geqq$28 | 7 (4.7%) |
| Diabetes mellitus, n (%) | |
| Present | 35(23.6) |
| Ca, mmol/L | 2.23 (2.10–2.34) |
| P, mmol/L | 1.74 (1.48–2.22) |
| iPTH, pg/ml | 308.50 (153.50–581.50) |
| Hb, g/L | 102.00 (95.25–110.00) |
| ALB, g/L | 38.25 (35.20–40.00) |
| sCR, μmol/L | 840.00 (666.00–910.00) |
| BUN, mmol/L | 19.50(18.10–21.08) |
| nPCR, g/kg/d | 0.81(0.75–0.86) |
| KT/V | 1.27(1.23–1.31) |
| FER, ng/ml | 193.80(129.63–220.00) |
| hs-CRP, mg/L | 1.11 (0.60–2.00) |
| Duration of HD, months | 42.00 (18.50–77.25) |

Notes: BMI, body mass index; Ca, serum calcium; P, serum phosphorus; iPTH, serum intact parathyroid hormone; Hb, Hemoglobin; ALB, serum albumin; sCR, serum creatinine; BUN, urea nitrogen; KT/V, urea nitrogen clearance index; nPCR, normalized protein catabolic rate; FER, ferritin; hs-CRP, serum hypersensitive C-reactive protein. Continuous variables with skewed data are summarized as the median (interquartile range); Dichotomous or categorical data are summarized as counts (percentages).

## Main results

**Degree of skin itching in patients with uremic pruritus as analyzed by the single-factor logistic-regression model.** Compared with patients without uremic pruritus, mild itching (OR = 1.917; $P$ = 0.010), moderate itching (OR = 2.696, $P$ < 0.001), and severe itching (OR = 6.915; $P$ < 0.001) were all related to hs-CRP level. The higher hs-CRP level was closely related to severe itching. iPTH was weakly associated with severe itching (OR = 1.001, $P$ = 0.031). Hb was slightly associated with moderate itching (OR = 1.036, $P$ = 0.026) and severe itching (OR = 1.036, P = 0.170), as shown in (Fig 2, S1 Table).

**Table 2. Demographic and clinical characteristics of patients with and without uremic pruritus.**

| Variable | Pruritic group (n = 60) | Non-pruritic group (n = 88) | P |
|---|---|---|---|
| Sex, n (%) | | | 0.318 |
| Male | 39 (65%) | 50 (56.8%) | |
| Age, n (%) | | | 0.083 |
| Non-elderly | 45 (75%) | 54 (61.4%) | |
| Educational level, n (%) | | | 0.623 |
| Illiterate or semi-literate | 7 (11.7%) | 16 (18.2%) | |
| Primary school | 13 (21.7%) | 23 (26.2%) | |
| Junior high school | 28 (46.7%) | 26 (29.5%) | |
| High school | 8 (13.3%) | 11 (12.5%) | |
| University and above | 4 (6.7%) | 12 (13.6%) | |
| Place of residence, n (%) | | | 0.602 |
| Rural | 42 (70%) | 58 (65.9%) | |
| BMI, n (%) | | | 0.079 |
| ≦18.4 | 13 (21.7%) | 12 (13.6%) | |
| 18.5–23.9 | 41 (68.3%) | 60 (68.2%) | |
| 24–27.9 | 5 (8.3%) | 10(11.4%) | |
| ≥28 | 1 (1.7%) | 6 (6.8%) | |
| Diabetes mellitus, n (%) | | | 0.099 |
| Present | 10 (16.7%) | 25 (28.4%) | |
| Ca, mmol/L | 2.24 (2.13–2.36) | 2.19 (2.09–2.34) | 0.211 |
| P, mmol/L | 1.85 (1.50–2.34) | 1.72 (1.44–2.15) | 0.117 |
| iPTH, pg/ml | 368.00 (229.75–754.25) | 220.50 (111.75–503.25) | 0.001* |
| Hb, g/L | 104.00 (98.25–111.75) | 100.00 (91.00–109.00) | 0.025* |
| ALB, g/L | 38.10 (35.13–40.00) | 38.45 (35.80–40.08) | 0.972 |
| sCR, µmol/L | 853.00 (686.00–950.00) | 824.00 (629.25–899.75) | 0.420 |
| BUN, mmol/L | 19.95 (18.28–21.65) | 19.10 (17.60–20.38) | 0.028* |
| nPCR, g/kg/d | 0.83(0.78–0.89) | 0.80(0.74–0.84) | 0.033* |
| KT/V | 1.27 (1.24–1.31) | 1.27 (1.22–1.32) | 0.567 |
| FER, ng/ml | 200.20 (127.58–218.25) | 190.50 (129.63–221.50) | 0.696 |
| hs-CRP, mg/L | 2.00 (1.00–3.00) | 0.90 (0.50–1.50) | <0.001* |
| Duration of HD, months | 45.50 (18.00–77.00) | 42.00 (20.25–78.75) | 0.936 |

Notes: The above numerical variables are all non-normal distribution data, articulated by the median (interquartile range). Intergroup comparisons of variables were made using the Mann–Whitney U test. Categorical variables are expressed as counts (percentages). Unordered categorical variables were tested by chi-square; ordered categorical variables including BMI and educational level were tested by Kruskal–Wallis *H*. BMI, body mass index; Ca, serum calcium; P, serum phosphorus; iPTH, serum intact parathyroid hormone; Hb, Hemoglobin; ALB, serum albumin; sCR, serum creatinine; BUN, urea nitrogen; nPCR, normalized protein catabolic rate; KT/V, urea nitrogen clearance index; FER, ferritin; hs-CRP, serum hypersensitive C-reactive protein.
*$P < 0.05$.

**Degree of skin itching in patients with uremic pruritus as analyzed by multivariate logistic regression.** We included four variables (hs-CRP, Hb, PTH, Place of residence) in multiple logistic regression analysis. Using no itching as a reference, the results showed that risk of skin itching increased with level of hs-CRP (mild skin itching, OR = 1.740; 95% CI, 1.061–2.845; $P = 0.028$; moderate skin itching, OR = 2.838; 95% CI, 1.744–4.618; $P < 0.001$; severe skin itching, OR = 9.440; 95% CI, 3.547–25.124; $P < 0.001$). Compared with urban

**Table 3. Baseline characteristics of patients with varying degrees of skin itching.**

| Variable | No itching (n = 88) | Mild itching (n = 22) | Moderate itching (n = 30) | Severe itching (n = 8) | P |
|---|---|---|---|---|---|
| Sex, n (%) | | | | | 0.495 |
| Male | 50 (56.8%) | 16 (72.7%) | 19 (63.3) | 4 (50%) | |
| Age, n (%) | | | | | 0.262 |
| Non-elderly | 54 (61. 4%) | 15 (68.2%) | 24 (80.0%) | 6 (75.0%) | |
| Educational level, n (%) | | | | | 0.478 |
| Illiterate or semi-literate | 16 (18.2%) | 2 (9.1%) | 4 (13.3%) | 1 (12.5%) | |
| Primary school | 23 (26.1) | 3 (13.6%) | 8 (26.7%) | 2 (25%) | |
| Junior high school | 26 (29.5%) | 11 (50.0%) | 12 (40.0%) | 5 (62.5%) | |
| High school | 11 (12.5%) | 2 (9.1%) | 6 (20.0%) | 0 (0.0%) | |
| University and above | 12 (13.6%) | 4 (18.2%) | 0 (0.0%) | 0 (0.0%) | |
| Place of residence, n (%) | | | | | 0.017* |
| Rural | 58 (65.9%) | 10 (45.5%) | 25 (83.3%) | 7 (87.5%) | |
| BMI, kg/m$^2$ | | | | | 0.300 |
| ≦18.4 | 12 (13.6%) | 4 (18.2%) | 6 (20.0%) | 3 (37.5%) | |
| 18.5–23.9 | 60 (68.2%) | 16 (72.8%) | 21 (70.0%) | 4 (50.0%) | |
| 24–27.9 | 10 (11.4%) | 1 (4.5%) | 3 (10.0%) | 1 (12.5%) | |
| ≥28 | 6 (6.8%) | 1 (4.5%) | 0 (0.0%) | 0 (0.0%) | |
| Diabetes mellitus, n (%) | | | | | 0.326 |
| Present | 25 (28.4%) | 4 (18.2%) | 4 (13.3%) | 2 (25%) | |
| Ca, mmol/L | 2.19 (2.09–2.34) | 2.25 (2.15–2.33) | 2.24 (2.12–2.45) | 2.17 (2.05–2.31) | 0.485 |
| P, mmol/L | 1.72 (1.44–2.15) | 1.90 (1.50, 2.47) | 1.90 (1.62, 2.25) | 1.49 (1.30–2.26) | 0.227 |
| iPTH, pg/ml | 220.50 (111.75–503.25) | 377.50 (280.00–719.00) | 360.00 (207.00–793.34) | 497.50 (229.50–801.25.) | 0.009* |
| Hb, g/L | 100.00 (91.00–109.00) | 103.50 (98.75–109.25) | 105.00 (98.75–109.25) | 107.00 (93.25–118.75) | 0.138 |
| ALB, g/L | 38.45 (35.80–40.08) | 38.05 (36.68–40.85) | 38.25 (34.73–39.63) | 36.60 (31.73–39.83) | 0.472 |
| sCR, μmol/L | 824.00 (629.25–899.75) | 853.00 (710.00–974.75) | 850.50 (709.50–955.00) | 788.50 (422.75–905.00) | 0.565 |
| BUN, mmol/L | 19.10 (17.60–20.38) | 19.90 (18.18–22.43) | 19.80 (18.18–21.20) | 20.35 (19.43–21.73) | 0.118 |
| nPCR, g/kg/d | 0.80(0.74–0.84) | 0.82(0.76–0.91) | 0.82(0.77–0.86) | 0.85(0.81–0.88) | 0.136 |
| KT/V | 1.27 (1.22–1.32) | 1.28 (1.26–1.33) | 1.27 (1.23–1.31) | 1.27 (1.22–1.33) | 0.708 |
| FER, ng/ml | 190.50 (129.63–221.50) | 203.00 (116.50–231.00) | 200.50 (142.26–206.00) | 153.75 (109.25–203.85) | 0.628 |
| hs-CRP, mg/L | 0.90 (0.50–1.50) | 1.08 (0.50–2.63) | 2.15 (1.18–2.93) | 3.93 (2.83–5.18) | <0.001* |
| HD duration, months | 42.00 (20.25–78.75) | 38.50 (17.25–83.25) | 37.00 (14.00–68.25) | 67.00 (56.00–89.25) | 0.119 |

Notes: P, Hb, ALB, Ca, PTH, sCR, hs-CRP, and HD duration are all non-normal data, articulated by the median (interquartile range); dichotomous or categorical data are summarized as counts (percentages). Kruskal–Wallis probability between the four groups, with subsequent multiple Mann–Whitney U test if significant. Categorical variables are expressed as counts (percentages), unordered categorical variables were tested by chi-square, and ordered categorical variables were tested by Kruskal–Wallis H. BMI, body mass index; Ca, serum calcium; P, serum phosphorus; iPTH, serum intact parathyroid hormone; Hb, Hemoglobin; ALB, serum albumin; sCR, serum creatinine; BUN, urea nitrogen; nPCR, normalized protein catabolic rate; KT/V, urea nitrogen clearance index; FER, ferritin; Hb, Hemoglobin; ALB, serum albumin; hs-CRP, serum hypersensitive C-reactive protein; HD, hemodialysis.

*P < 0.05.

residents, rural residents have a higher risk of moderate itching (OR = 3.869; 95% CI, 1.099–13.622; P = 0.035). as shown in (Fig 3, S2 Table).

## Other analyses

The level of iPTH in patients with mild itching was higher than in those with no skin itching, to a statistically significant degree (P <0.05). Levels of hs-CRP in patients with moderate and severe skin itching were statistically significantly higher than in patients without itching (all P <0.001).

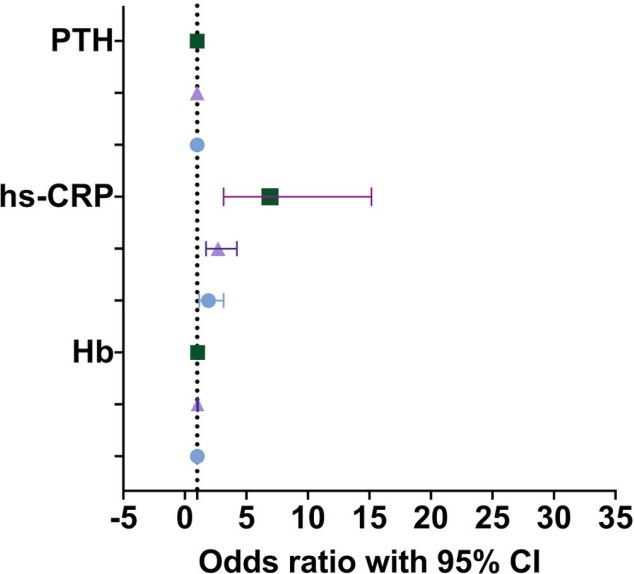

**Fig 2. Single-factor logistic-regression analysis of the degree of skin itching in patients with uremic pruritus.** The error bars represent the lower and upper limits of the 95% confidence interval. The symbol is a circle for mild itching, a triangle for moderate itching, and a square for severe itching. The dashed line in the figure indicates that the odds ratio is equal to 1.

There was no statistically significant difference in place of residence in patients with mild, moderate, and severe itching compared with no skin itching (all *P* > 0.05), As shown in Fig 4.

## Discussion

### Key results

In this study, we found that among 148 eligible patients receiving maintenance HD, 60 had UP (prevalence rate, 40.54%); they included 22 mild cases of skin itching (14.86%), 30 moderate

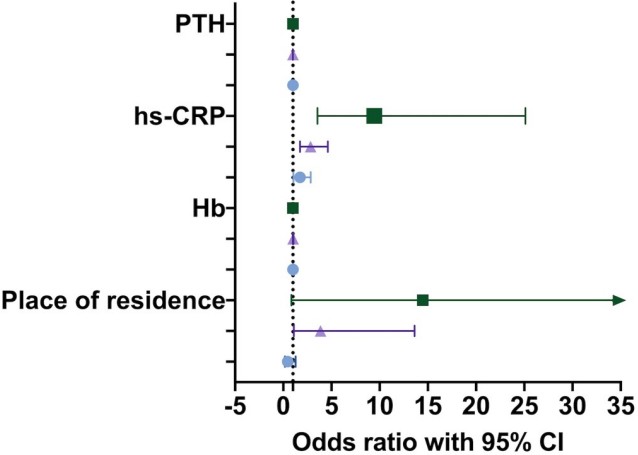

**Fig 3. Multiple logistic regression analysis of factors influencing skin itching in patients with uremic pruritus.** The error bars represent the lower and upper limits of the 95% confidence interval. The symbol is a circle for mild itching, a triangle for moderate itching, and a square for severe itching. The dashed line in the figure indicates that the odds ratio is equal to 1.

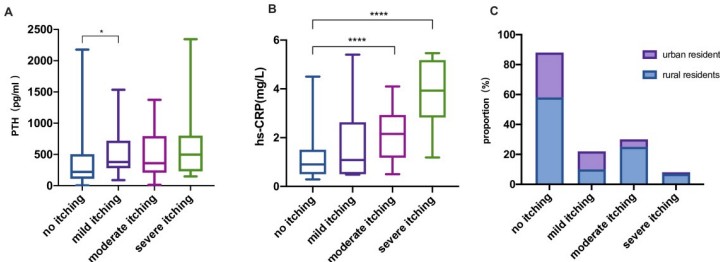

**Fig 4. Comparison of patients with varying degrees of skin itching and varying hs-CRP and IPTH levels.** (A) Intergroup comparison of iPTH levels; (B) intergroup comparison of hs-CRP levels; (C) intergroup comparison of the composition ratio of rural to urban residents. $****P < 0.0001$, $*P < 0.05$. Error line represents the minimum and maximum values.

cases (20.27%), and 8 severe cases (5.41%). The composition ratio of rural to urban patients, levels of IPTH and hs-CRP were statistically significantly different in patients with varying degrees of skin pruritus. Single-factor logistic-regression analysis discovered a link between hs-CRP, Hb and iPTH and varying degrees of itching. Multifactor logistic regression analysis indicated that patients with higher hs-CRP levels had higher risks of mild, moderate, and severe skin itching. Interestingly, the multivariate analysis showed that rural residents have a higher risk of moderate skin itching than urban residents.

### Limitations

Our study had some limitations. Firstly, it was a single-center, cross-sectional study that did not show a causal relationship between severity of skin itching in UP and hs-CRP, so further validation is needed from multicenter, large-sample, prospective cohort studies. Secondly, this study did not include some of the influences known in the literature such as diet, insulin resistance, interleukins 6 and 2, and κ opioid receptor distribution. Thirdly, the number of patients with severe pruritus is small (n = 8), so readers may not understand a meaningful association with severe pruritus. Fourthly, the sample size of this study was limited and uneven between groups. Fifthly, this study recruited 160 patients, 3 of whom were non-responders, and these non-responders may have differed from responders in some critical clinical features or exposures; however, the proportion (1.87%) was small. Finally, during participation in the questionnaire of this study, some patients may be limited by their cultural level or subjective perceptions, which may lead to discrimination in the individual skin itch scale scores, thus affecting the accuracy of the assessment of the itchiness of the patient's skin. For the study participants who took part in the questionnaire, we trained two research staff, and the final results were taken as the mean of the two-pruritus skin itch scale scores.

### Interpretation

This study showed that median hs-CRP level gradually increased along with the degree of skin itching in patients with uremic pruritus (no itching, 0.90; mild skin itching, 1.08; moderate skin itching, 2.15; severe skin itching, 3.93). Furthermore, hs-CRP levels in patients with moderate and severe skin itching were significantly higher than in those without itching. These findings suggested that hs-CRP levels might be associated with the severity of skin itching. Jiang et al. [8] reported that levels of C-reactive protein (CRP) and leukocyte interleukin-6 (IL-6) in patients receiving high-flux hemodialysis improved, compared with those in patients who received hemodialysis filtration. Another study also showed that HD patients with UP

had significantly higher serum CRP and IL-6 levels compared to HD patients without UP, UP seems to be associated with an up-regulation of micro-inflammation in uremia [9]. Mahmud-pour's team [10] indicated that the average reduction in VAS scores in the montelukast group was greater than that in the placebo group, and average levels of hs-CRP in this group showed a downward trend. However, average hs-CRP levels in the placebo group gradually increased, showing that the effect of montelukast treatment in UP might be related to a reduction in an inflammatory response. A factor analysis also shows a close correlation between UP and levels of the inflammatory factors CRP and leukocyte IL-6 [11]. Based on this study and previous studies, inflammatory responses appear to play an important role in UP.

We analyzed the factors affecting the intensity of skin itching in UP patients and corrected for patients' levels of Hb and iPTH, Place of residence. Multifactor logistic-regression analysis found that hs-CRP levels were associated with the intensity of skin itching and were a risk factor. Compared with patients without itching, the OR value for mild skin itching was 1.740, that for moderate skin itching was 2.838, and that for severe skin itching was 9.440, indicating that degree of skin itching increased along with hs-CRP levels. A previous multivariable logistic-regression analysis confirmed that older age and higher CRP levels are associated with severe itching [3]. Still, in our study, neither single-factor nor multifactor logistic-regression analysis showed any differences in skin itching between elderly and non-elderly patients. In our study, the OR value for hs-CRP of patients with severe skin itching was significantly higher than that of patients without itching, suggesting that higher hs-CRP levels were closely associated with severe skin itching, which was consistent with the results of an earlier study. Pakfetrat et al. [12] assessed the effect of sertraline on UP. Patients were randomly divided into two groups, the trial group receiving sertraline and the control group taking a placebo. Their severity of itching was measured by two scoring systems (visual-simulation score and DUO scoring system [13]). The researchers found that differences in skin itching intensity between the two scoring systems and between the two groups were directly related to CRP level, and they believed that sertraline reduced skin itching, perhaps due to its effect of reducing the number of inflammatory cytokines. Tseng et al. [14] studied a possible association between vegetarians and UP. They found that levels of inflammatory factors such as leukocyte IL-2 and hs-CRP, visual-simulation score (Analog Score and VAS), and itching score were all lower in vegetarian patients than in non-vegetarian ones. This study demonstrated that a vegetarian diet might be associated with improvement of uremic-pruritus severity in hemodialysis patients.

In our study, the composition ratio showed a significant difference between rural and urban patients with different degrees of skin itching. Although the single factor analysis did not indicate that the place of residence was related to the intensity of the patient's skin itching, the multifactor analysis demonstrated that rural residents had a higher risk of moderate skin itching than urban residents. UP in patients receiving maintenance HD is associated with air pollutants such as $NO_2$ and CO [15]. A cross-sectional study also suggested that UP in patients receiving maintenance HD might be associated with the number of days in which environmental particulate matter $\leq 2.5$ μ in diameter (PM2.5) exceeds standard levels [16]. The above mentioned two studies suggest that itching of the skin might be associated with the patient's living environment. In this study, there were a total of 148 patients with uremia and eight patients with severe skin pruritus, accounting for 5.4% of the total patients. 2012–2015 DPPSO study [3] surveyed 7,629 people, divided into five groups (not at all, somewhat, moderately, Very Much, Extremely) according to the degree of itchy skin. Extreme itching accounted for 7%, which is slightly higher than 5.4% of patients with severe itching in this study. The reason for the differences may lie in the different criteria used to assess the degree of pruritus in the patient's skin.

iPTH levels differed between patients with varying degrees of skin itching. Median iPTH level in patients with severe skin itching was higher than that in patients without itching. Still,

there was no statistically significant difference between iPTH levels across groups, and single-factor regression analysis suggested a weak association between higher iPTH levels and severe skin itching. However, after we corrected for hs-CRP and Hb levels, place of residence, that association disappeared. After parathyroid excision, iPTH level and serum phosphate concentration are significantly reduced, and skin itching disappears [17, 18]. A study by Moldovan et al. [19] suggested that the average value of PTH decreased significantly after parathyroidectomy. However, itching of the skin did not improve. Levy et al. [20] studied the associated symptoms of secondary hyperparathyroidism in patients receiving maintenance HD. Results showed that an increase in PTH levels was associated with skin itching worsening over time, and the effect was more obvious with larger changes in PTH level. However, the combination of the disease, detection of medication, the inclusion of up to 19 symptoms, the larger *P*-value threshold (0.1), and other promiscuous factors might have affected the reliability of that study. It is therefore controversial whether there is a link between iPTH and the degree of itchiness of the skin.

Some studies suggested that UP is related to Ca, P, ALB, and uremic toxins [12, 21]. Nevertheless, contradictory data on the impact of Ca and P have been reported [22]. In this study, no statistically significant differences were found in Ca, P, ALB, and creatinine when comparing groups of patients with different degrees of skin pruritus. Besides, single-factor logistic regression analysis also did not find an association between these factors and the degree of skin pruritus. Therefore, these factors were ultimately not included in the multifactor logistic regression analysis. In fact, despite various substances have been discussed as potential pruritogens in chronic renal diseases, a causal relationship could never be established [23].

At present, only a few studies have explored the factors affecting skin itching intensity in these patients. Wieczorek et al. [24] documented Intensity of skin itching have a significant negative correlation with expression of κ opioid receptors in patients with HD. Also, hs-CRP levels, male sex, HD duration, insulin resistance, insufficient dialysis adequacy, and hyperphosphatemia are positively correlated with UP intensity [25] and increasing the blood flow of the HD machine can reduce the intensity and frequency of skin itching [26]. However, the above preliminary research used visual-analogue scoring (VAS) to evaluate the degree of skin itching. In this study, we used a five-dimensional itching scale to evaluate skin itching across multiple dimensions in uremic patients undergoing maintenance HD. Although VAS is sufficient to assess the severity of symptoms, it does not take into account other aspects of pruritus, such as the relative impact of pruritus on quality of life, changes in pruritus over time, and the location of pruritus. The 5-D score was strongly correlated with the numerical rating scale (NRS) and VAS [5, 27], which is a questionnaire with acceptable validity, reliability and sensitivity to change to evaluate pruritus in Thai and Arabic patients [27, 28]. The 5-dimensional Pruritus Scale is recommended to assess the severity of pruritus because it is more accurate than VAS and more sensitive to the multidimensional nature of pruritus [29].

## Generalizability

In conclusion, our study suggested that the intensity of skin itching in patients with UP was associated with hs-CRP and that higher hs-CRP levels were associated with severe skin itching. Besides, the environment may be related to the degree of itchy skin in uremia patients.

## Supporting information

**S1 Table. Single-factor logistic-regression analysis of the degree of skin itching in patients with uremic pruritus.**
(DOCX)

**S2 Table. Multiple logistic regression analysis of factors influencing skin itching in patients with uremic pruritus.**
(DOCX)

## Acknowledgments

We thank LetPub (www.letpub.com) for its linguistic assistance during the preparation of this manuscript.

## Author Contributions

**Conceptualization:** Jian-Hui Zhao, Yi-Wen Li.

**Data curation:** Qiu-Shuang Zhu, Li-Li Wang.

**Investigation:** Jian-Hui Zhao, Qiu-Shuang Zhu, Yi-Wen Li.

**Methodology:** Jian-Hui Zhao, Yi-Wen Li.

**Project administration:** Jian-Hui Zhao.

**Supervision:** Yi-Wen Li.

**Writing – original draft:** Jian-Hui Zhao.

**Writing – review & editing:** Jian-Hui Zhao, Qiu-Shuang Zhu, Yi-Wen Li.

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
