## [Decision Letter · Decision Letter 0]

24 Sep 2020

PONE-D-20-18098

Determinants of the intensity of uremic pruritus in patients receiving maintenance hemodialysis: a corss-sectional study

PLOS ONE

Dear Dr. Zhao,

Thank you for submitting your manuscript to PLOS ONE. After careful consideration, we feel that it has merit but does not fully meet PLOS ONE’s publication criteria as it currently stands. Therefore, we invite you to submit a revised version of the manuscript that addresses the points raised during the review process.

We look forward to receiving your revised manuscript.

Kind regards,

Girish Chandra Bhatt, MD, FASN

Academic Editor

PLOS ONE

Journal Requirements:

Reviewers' comments:

Reviewer's Responses to Questions

**Comments to the Author**

1. Is the manuscript technically sound, and do the data support the conclusions?

Reviewer #1: No

Reviewer #2: Yes

2. Has the statistical analysis been performed appropriately and rigorously? 

Reviewer #1: I Don't Know

Reviewer #2: Yes

3. Have the authors made all data underlying the findings in their manuscript fully available?

Reviewer #1: Yes

Reviewer #2: Yes

4. Is the manuscript presented in an intelligible fashion and written in standard English?

Reviewer #1: Yes

Reviewer #2: Yes

5. Review Comments to the Author

Reviewer #1: The authors performed a cross-sectional analysis of 148 hemodialysis patients to identify factors associated with uremic pruritus. They found that higher levels of PTH, hemoglobin, and hs-CRP as well as rural residence of patients were associated with worse pruritus. There are major concerns with the lack of dialysis and other pertinent laboratory data. Overall, the results do not add new data to the existing literature.

Comments:

1) There is no data provided for dialysis adequacy, which has been shown to be associated with pruritus in prior studies. Likewise, BUN levels, nPCR, ferritin levels, and diabetic status should be included in the analysis.

2) The authors used the Five-Dimensional Itching Scale (5DIS) to assess the degree of pruritus in patients, noting that this scale would “produce more objective and comprehensive results” compared to the visual-analogue scoring scale. The authors should provide data to justify this statement.

3) A large proportion of the discussion was dedicated to the finding that higher hs-CRP levels were associated with pruritus. A prior study has previously reported this finding, along with levels of cytokines and T cells (PMID 16249205); this reference should be cited.

4) The number of patients with severe pruritus is small (n=8) and therefore meaningful associations with severe pruritus may not be appreciated.

Reviewer #2: In this study, the authors studied uremic pruritus using the five dimensional itch scale in 148 adult patients of end-stage renal disease receiving haemodialysis and report prevalence of 40.5% and association with parathormone (PTH), hemoglobin and high-sensitive C reactive protein (hs-CRP) on bivariate analysis and hs-CRP alone on multiple logistic regression. The study seems to be designed logically, conducted scientifically and analyzed with due statistical rigour. However, the manuscript is lengthy, does not have a smooth flow of information and requires rewriting as per the STROBE checklist. Major observations are described below. Minor observations are inserted in the uploaded version of the manuscript as “sticky notes” at appropriate places.

Major observations

1. The introduction should succinctly summarize the present state of knowledge on uremic pruritus, highlight the gap in knowledge the study sought to address and state the study’s objective. Comparison with previous studies (measurement of severity of itching) should be reserved for discussion.

2. The stated aims of the study are to “1. explore the prevalence of UP in maintenance HD patients and the demographic and clinical characteristics of pruritus patients.

2.explore determinants of higher pruritus intensity based on participants’ clinical and biochemical parameters and their pruritus assessments by 5DIS at baseline”. The second aim is somewhat unclear. Were characteristics of patients with higher pruritus intensity to be compared with those of patients with of lower pruritus intensity or with those of patients without pruritus? Clear statement in the PICO format is suggested. What is the novelty of the research question? This needs to be justified.

3. The setting of the study may be described in “Methods”.

4. How were biochemical parameters such as serum calcium, phosphorus, creatinine, etc assayed? The detailed methods need not be described but should be referenced.

5. The study reports no significant association of uremic pruritus with factors such as calcium, phosphorus, creatinine and albumin. However, the sample size calculation was made to estimate the prevalence of uremic pruritus. Could the study have been underpowered for the study of these factors?

6. The following factors influencing uremic pruritus have not been studied: diet, insulin resistance, dialysis adequacy, interleukins 6 & 2, and κ opioid receptor distribution. These need to be acknowledged as limitations.

6. PLOS authors have the option to publish the peer review history of their article (what does this mean?). If published, this will include your full peer review and any attached files.

Reviewer #1: No

Reviewer #2: No

---

## [Author Response · Author response to Decision Letter 0]

21 Oct 2020

Response to the PLOS ONE’s editor

Thank you for the hard work of the academic editors of “PLOS ONE”. Thank you for accompanying me and helping me grow up. As for the issues that need my attention, I summarize as follows. 

Response：

We have revised the paper following your journal's formatting requirements for title, authorship, and text.

2.While revising your submission, please upload your figure files to the Preflight Analysis and Conversion Engine (PACE) digital diagnostic tool, https://pacev2.apexcovantage.com/. PACE helps ensure that figures meet PLOS requirements. To use PACE, you must first register as a user. Registration is free. Then, login and navigate to the UPLOAD tab, where you will find detailed instructions on how to use the tool. If you encounter any issues or have any questions when using PACE, please email PLOS at figures@plos.org. Please note that Supporting Information files do not need this step. 

Response：

A total of four images are in the paper, and the image files have been uploaded to the PACE software, all with a resolution of 600ppi, 2000-4500 pixels wide and 1358-3925 pixels high, and are currently sealed in the PACE software.

Response to Reviewer

Thank the reviewers for these lovely comments concerning our manuscript entitled “Determinants of the intensity of uremic pruritus in patients receiving maintenance hemodialysis: A cross-sectional study”. These comments are all valuable and very helpful for revising and improving our paper, as well as the essential guiding significance to our researches. We have studied comments carefully and have made corrections which we hope to meet with approval. The response to the reviewer’s comments are as follows: 

Reviewer #1 

The authors performed a cross-sectional analysis of 148 hemodialysis patients to identify factors associated with uremic pruritus. They found that higher levels of PTH, hemoglobin, and hs-CRP, as well as the rural residence of patients were associated with worse pruritus. There are major concerns with the lack of dialysis and other pertinent laboratory data. Overall, the results do not add new data to the existing literature. 

1. There is no data provided for dialysis adequacy, which has been shown to be associated with pruritus in prior studies. Likewise, BUN levels, nPCR, ferritin levels, and diabetic status should be included in the analysis. 

Response:

The revision adds these factors, such as urea nitrogen, ferritin level, diabetes, and single pool KT/V(spKT/V), to the list of factors related to hemodialysis. Normalized protein catabolic rate (nPCR) is a nutritional index for hemodialysis patients. Its measurement requires multiple blood collections, which can be challenging to obtain, especially with the inclusion of serum albumin, which is a nutritional index. The spKT/V index reflects the adequacy of single hemodialysis. 

This section is mainly shown on pages 6-11, tables 1-3.

2. The authors used the Five-Dimensional Itching Scale (5DIS) to assess the degree of pruritus in patients, noting that this scale would “produce more objective and comprehensive results” compared to the visual-analogue scoring scale. The authors should provide data to justify this statement. 

Response:

In this study, we used a five-dimensional itching scale to evaluate skin itching across multiple dimensions in uremic patients undergoing maintenance HD. Although VAS is sufficient to assess the severity of symptoms, it does not take into account other aspects of pruritus, such as the relative impact of pruritus on quality of life, changes in pruritus over time, and the location of pruritus. The 5-D score was strongly correlated with the numerical rating scale（NRS）and VAS[5, 26], which is a questionnaire with acceptable validity, reliability and sensitivity to change to evaluate pruritus in Thai and Arabic patients[26, 27]. The 5-dimensional Pruritus Scale is recommended to assess the severity of pruritus because it is more accurate than VAS and more sensitive to the multidimensional nature of pruritus[28]. 

This section is detailed on page15, line 29-38.

3. A large proportion of the discussion was dedicated to the finding that higher hs-CRP levels were associated with pruritus. A prior study has previously reported this finding, along with levels of cytokines and T cells (PMID 16249205); this reference should be cited.

Response：

This section is detailed on Reference #8（page 13, line35）

4. The number of patients with severe pruritus is small (n=8) and therefore, meaningful associations with severe pruritus may not be appreciated. 

Response：

In this study, there were a total of 148 patients with uremia and eight patients with severe skin pruritus, accounting for 5.4% of the total patients. 2012-2015 DPPSO study[3] surveyed 7,629 people, divided into five groups (not at all, somewhat, moderately, Very Much, Extremely) according to the degree of itchy skin. Extreme Itching accounted for 7%, which is slightly higher than 5.4% of patients with severe Itching in this study. The reason for the differences may lie in the different criteria used to assess the degree of pruritus in the patient's skin.

This section is detailed on page 14, lines 37-43

Reviewer #2 

In this study, the authors studied uremic pruritus using the five-dimensional itch scale in 148 adult patients of end-stage renal disease receiving hemodialysis and report the prevalence of 40.5% and association with parathormone (PTH), hemoglobin and high-sensitive C reactive protein (hs-CRP) on bivariate analysis and hs-CRP alone on multiple logistic regression. The study seems to be designed logically, conducted scientifically and analyzed with due statistical rigour. However, the manuscript is lengthy, does not have a smooth flow of information and requires rewriting as per the STROBE checklist. Major observations are described below. Minor observations are inserted in the uploaded version of the manuscript as “sticky notes” at appropriate places. 

Response:

 1）We have rewritten the paper according to the STROBE checklist, details of which can be found in the attached STROBE checklist.

 2）Concerning some minor observation, we have followed your comments with deep appreciation.

Major observation:

1. The introduction should briefly summarize the present state of knowledge on uremic pruritus, highlight the gap in understanding the study sought to address and state the study’s objective. Comparison with previous studies (measurement of severity of Itching) should be reserved for discussion.

Response: 

The reviewer's suggestions were excellent, and we have corrected them.

This section has been moved to page 15, lines 23-29.

2. The stated aims of the study are to “1. explore the prevalence of UP in maintenance HD patients and the demographic and clinical characteristics of pruritus patients. Explore determinants of higher pruritus intensity based on participants’ clinical and biochemical parameters and their pruritus assessments by 5DIS at baseline”. The second aim is somewhat unclear. Were characteristics of patients with higher pruritus intensity to be compared with those of patients with of lower pruritus intensity or with those of patients without pruritus? Clear statement in the PICO format is suggested. What is the novelty of the research question? This needs to be justified.

Response: 

objectives

(1). explore the prevalence of UP in maintenance HD patients and the demographic and clinical characteristics of pruritus patients. 

(2). analyze the relationship between high-sensitivity C-reactive protein and different degrees of skin itching.

This section has been moved to page 2, lines 28-32.

3. The setting of the study may be described in “Methods”. 

Response:

This section is detailed on Methods (study setting), page2, lines 38 & page3, lines 1-5. 

4.How were biochemical parameters such as serum calcium, phosphorus, creatinine, etc assayed? The detailed methods need not be described but should be referenced. 

Response:

We assayed biochemical indicators such as Ca, P, iPTH, Hb, ALB, sCr, Bun, FER, sp/KTV and hs-CRP. The hospital's central laboratory performed all of the laboratory tests, and auto-analyzers were used to determine biochemical data, iPTH was measured with Roche second-generation assay. The clinical biochemical indicators are based on the data within three months from the survey day. If there is no data within three months, the patient is required to re-test. If the patient is unwilling to cooperate with the test, it is a missing value. spKT/V（single pool KT/V）=-ln⁡(R-0.008t)+(4-3.5R)*(∆BW/BW). Note: K refers to Blood urea clearance rate of dialyzer（L/h）, t is the dialysis time (h), and V is the distribution volume of urea (V). R is the ratio of blood urea nitrogen after dialysis to blood urea nitrogen before dialysis; t is single dialysis time in h; ∆BW is the weight change value before and after dialysis, i.e. ultrafiltration, unit L; BW is weight in kg. Blood collection requirements: blood samples before dialysis from the artery end of the vascular path, after dialysis before blood sample collection to stop ultrafiltration, reduce blood flow of 50 ml/min, wait 15 seconds after blood collection from the artery as a blood sample after dialysis. 

This section is detailed on page 3, lines 36-41 & page 4, lines 1-7.

5.The study reports no significant association of uremic pruritus with factors such as calcium, phosphorus, creatinine and albumin. However, the sample size calculation was made to estimate the prevalence of uremic pruritus. Could the study have been under powered for the study of these factors? 

Response:

Some studies suggested that UP is related to Ca, P, ALB, and uremic toxins[11, 20]. Nevertheless, contradictory data on the impact of Ca and P have been reported[21]. In this study, no statistically significant differences were found in Ca, P, ALB, and creatinine when comparing groups of patients with different degrees of skin pruritus. There may indeed be a lack of test power（Since the data for these factors are non-normally distributed, the exact distribution is unknown. Based on statistical theory, it is not possible to calculate the true test power with certainty.）, which is considered to be related to the small sample size of this study and the unbalanced sample size among the groups. However, single-factor logistic regression analysis also did not find an association between these factors and the degree of skin pruritus. Therefore, these factors were ultimately not included in the multifactor logistic regression analysis. In fact, despite various substances have been discussed as potential pruritogens in chronic renal diseases, a causal relationship could never be established[22]. 

This section is detailed on page 15, lines 14-22.

6.The following factors influencing uremic pruritus have not been studied: diet, insulin resistance, dialysis adequacy, interleukins 6 & 2, and κ opioid receptor distribution. These need to be acknowledged as limitations. 

Response:

Firstly, this study was a single-center study with a limited number of people on hemodialysis, and therefore the number of people included in the study was limited. Many factors influence skin itching in uremic patients, and it is not practical to include all relevant aspects of the study. Secondly, there is the issue of patient compliance. For example, to study diet factor, we need to distinguish between vegetarian and non-vegan patients, which requires strict control of the patient's diet. For insulin resistance, if the HOMA model is used to assess the resistance of the body to insulin (HOMA-IR), fasting blood glucose and fasting insulin levels need to be measured based on the formula HOMA-IR=FPG×FINS/22.5, therefor, fasting blood needs to be collected before hemodialysis. Some patients are difficult to cooperate, the missing value of the study will be further increased. Thirdly, for dialysis adequacy, we included a urea clearance index (kt/v) to assess this. In conclusion, it is essential to acknowledge the limitations of this study.

This section is detailed on page 13, lines 17-19.

---

## [Decision Letter · Decision Letter 1]

20 Nov 2020

PONE-D-20-18098R1

Determinants of the intensity of uremic pruritus in patients receiving maintenance hemodialysis: A cross-sectional study

PLOS ONE

Dear Dr. Zhao,

Thank you for submitting your manuscript to PLOS ONE. After careful consideration, we feel that it has merit but does not fully meet PLOS ONE’s publication criteria as it currently stands. Therefore, we invite you to submit a revised version of the manuscript that addresses the points raised during the review process.

We look forward to receiving your revised manuscript.

Kind regards,

Girish Chandra Bhatt, MD, FASN

Academic Editor

PLOS ONE

Reviewers' comments:

Reviewer's Responses to Questions

**Comments to the Author**

1. If the authors have adequately addressed your comments raised in a previous round of review and you feel that this manuscript is now acceptable for publication, you may indicate that here to bypass the “Comments to the Author” section, enter your conflict of interest statement in the “Confidential to Editor” section, and submit your "Accept" recommendation.

Reviewer #1: All comments have been addressed

Reviewer #2: All comments have been addressed

2. Is the manuscript technically sound, and do the data support the conclusions?

Reviewer #1: Yes

Reviewer #2: Yes

3. Has the statistical analysis been performed appropriately and rigorously? 

Reviewer #1: I Don't Know

Reviewer #2: Yes

4. Have the authors made all data underlying the findings in their manuscript fully available?

Reviewer #1: Yes

Reviewer #2: Yes

5. Is the manuscript presented in an intelligible fashion and written in standard English?

Reviewer #1: Yes

Reviewer #2: Yes

6. Review Comments to the Author

Reviewer #1: The authors have revised the manuscript in response to the initial comments. Improvements include addition of other variables that have been associated with pruritus into their analysis along with data to support their proposed use of the 5D score to assess pruritus. Overall, the manuscript adds to existing reports of hsCRP’s association with uremic pruritus. One minor comment to consider is that the nPCR can be obtained from the kinetic modeling used to obtain spKt/V. But since the BUN levels were not different across pruritus severity, it is unlikely that the nPCR will be different.

Reviewer #2: The authors have rewritten the manuscript and it reads much more clearly. Small changes are suggested. These have been indicated as "sticky notes" at the appropriate places in the uploaded version of the manuscript.

7. PLOS authors have the option to publish the peer review history of their article (what does this mean?). If published, this will include your full peer review and any attached files.

Reviewer #1: No

Reviewer #2: No

---

## [Author Response · Author response to Decision Letter 1]

3 Dec 2020

Dear Dr Girish Chandra Bhatt, MD, FASN and Reviewers

Thank you for giving us the opportunity to submit a revised draft of the manuscript “Determinants of the intensity of uremic pruritus in patients receiving maintenance hemodialysis: A cross-sectional study” for publication in the Journal of PLOS ONE. 

We have incorporated all of the suggestions made by the reviewers. Those changes are highlighted in the revised manuscript with tracked changes. Please see a point-by-point response to the reviewers’ comments and concerns. Page numbers and line numbers in the original manuscript are shown in black and bold; page numbers and line numbers in revised manuscript with tracked changes are shown in normal type.

We hope that the revised manuscript is acceptable manuscript for publication in your Journal.

Sincerely yours,

Jian-Hui Zhao

Response to academic editor

1.Response: we would not like to make changes to our financial disclosure.

2.If applicable, we recommend that you deposit your laboratory protocols in protocols.io to enhance the reproducibility of your results. Protocols.io assigns your protocol its own identifier (DOI) so that it can be cited independently in the future. For instructions see: http://journals.plos.org/plosone/s/submission-guidelines#loc-laboratory-protocols

Response: Our study is a clinical cross-sectional survey study and does not involve complex laboratory procedures and clinical research methods. However, we have found a simple formula for calculating normalized protein catabolic rate (nPCR). In this formula, as long as there is urea nitrogen before hemodialysis and the KT/V result of single hemodialysis, nPCR can be calculated accurately. We deposit this method on the protocols.io website to get the doi number (https://dx.doi.org/10.17504/protocols.io.bp94mr8w) [PROTOCOL DOI].

3.While revising your submission, please upload your figure files to the Preflight Analysis and Conversion Engine (PACE) digital diagnostic tool, https://pacev2.apexcovantage.com/. PACE helps ensure that figures meet PLOS requirements. To use PACE, you must first register as a user. Registration is free. Then, login and navigate to the UPLOAD tab, where you will find detailed instructions on how to use the tool. If you encounter any issues or have any questions when using PACE, please email PLOS at figures@plos.org. Please note that Supporting Information files do not need this step. 

Response: We have updated Fig 1, but Fig 2-4 do not require any changes. PACE can ensure that these figures are PLOS compliant.

Response to Reviewer

Reviewer #1 

The authors have revised the manuscript in response to the initial comments. Improvements include addition of other variables that have been associated with pruritus into their analysis along with data to support their proposed use of the 5D score to assess pruritus. Overall, the manuscript adds to existing reports of hsCRP’s association with uremic pruritus. One minor comment to consider is that the nPCR can be obtained from the kinetic modeling used to obtain spKt/V. But since the BUN levels were not different across pruritus severity, it is unlikely that the nPCR will be different.

Response: 

Thank you for your suggestions on the data integrity and statistical analysis aspects of your paper, we see that you have given us a kind reminder in your comments. Here are the answers to your concerns.

Following your suggestion, we have added a variable (nPCR) in the revised manuscript, based on pre-dialysis blood urea nitrogen levels in combination with spKT/V, to introduce the formula: nPCR((g/kg)/d)=C_0/[a+b×KT/V+c÷(KT/V)] +0.168,where C0 indicates pre-dialysis blood urea nitrogen in mg/dl (1mmol/L urea nitrogen is equal to 2.802mg/dl). a, b, c has different coefficients depending on the time of the dialysis schedule. In our blood purification center, patients are dialyzed three times a week, either on Mondays, Wednesdays, Fridays or either Tuesdays, Thursdays, Saturdays, so that the time between the first dialysis sessions at the beginning of the week is longer. The following formula is used: beginning-of-week: nPCR(g/kg/d)=C_0/[36.3+5.48×KT/V+53.5÷(KT/V)] +0.168. 

Our statistical analysis showed that patients with pruritus have a higher nPCR level compared to patients without pruritus (p=0.028), however, there is no statistically significant difference between patients with different levels of pruritus (p=0.136). Furthermore, the single factor analysis also did not show a statistically significant correlation, thus the results of the multifactorial analysis did not change and the conclusions of the paper were unchanged. 

This section was shown in lines 7-13, page 4 in revised manuscript with tracked changes and the detailed data are shown in Tables 1-3. 

Reviewer #2 

The authors have rewritten the manuscript and it reads much more clearly. Small changes are suggested. These have been indicated as "sticky notes" at the appropriate places in the uploaded version of the manuscript.

Response: 

We appreciate your suggestions regarding the language structure of our paper, which is very conducive to improving the quality of the paper, and the following is an itemized response to your small comments. 

1. Line 25, page 2: diabetes mellitus (avoid unnecessary capitalization)

This phrase was modified according to the comment. (Line 24, page 2). 

2. Line 29, page 2: Objectives (headings should be capitalized)

Thank you for pointing this out. As you suggested, we have corrected the “objectives” to “Objectives”. (Line 28, page 2). 

3. Line 30 page 3: we feel sorry for our carelessness in our resubmitted manuscript, the error is revised. 

 We have corrected “BMI” to “Body Mass Index (BMI)”. (Line 27, page 3). 

4. Line 31 page 3: 

We have corrected “normal patients with BMI” to “patients with normal BMI”. (Line 28, page 3). 

5. line 36 page 3: 

Thank you for pointing this out. As you suggested, we have corrected the “Place of residence “to “place of residence”. (Line 33, page 3). 

6. line 39 page 3: Avoid using the term "such as".

“such as “is replaced with “comprised of”. (line 36 page 3)

7. line 4 page 4: 

Thank your suggestions, we have corrected it. (line 41 page 3)

8. line 12 page 4: The section on bias may be included into the discussion as part of limitations of the study. 

The section on bias was included into the discussion as part of limitations of the study. (lines 19-27, page13).

9. line 27 page 4

This phrase was modified according to the comment. (Line 14, page 4). 

10. line 30 page 4 

This phrase was modified according to the comment. (Line 17, page 4). 

11. line 34 page 4: The section on "quantitative variables" may be included in "statistical methods". This may be rewritten with a more logical flow. 

The section on "quantitative variables" have been included in "statistical methods" according to your suggestions and we also rewritten this section. (lines 22-37 page 4)

12. line 9 page 5: Is it 4 missing values in two patients and therefore a total of 8 missing values?

Respond: yes.

13. line 14 page 5: There were a total of 180 patients. Why were 20 patients excluded. This should be clarified. 

Our blood Purification Center has a total of 180 patients. 20 patients were excluded, and 10 of them were unwilling to participate in the survey. 5 were peritoneal dialysis combined with hemodialysis, but hemodialysis was only treated once a week, and the remaining 5 were scheduled for 2 dialysis a week. Therefore, a total of 160 patients who regularly underwent hemodialysis three times a week were included in this study from March 2019 to June 2019. (line40 page 4 & lines 1-5 page 5). Besides, we updated Fig 1.

14. Duplication should be avoided. For instance, if the number of males is shown, the number of females need not be shown. 

We have modified these phrase, See Table 1 for details. 

15. The values are shown as n (%) or median (IQR). This should be specified in the table. 

Continuous variables with skewed data are summarized as the median (interquartile range); Dichotomous or categorical data are summarized as counts (percentages). 

As shown in the section on notes in Table 1 for details. 

16. Line 2 page 13: This section of key result in discussion may be rewritten with brevity. 

The section has been rewritten and streamlined. (lines3-11 page 13).

17. Line 22 page 13: This section may be rewritten with brevity. 

The section has been rewritten and streamlined. (lines 29-40 page 13 & lines1-44 page 14 & lines 1-35 page 15).

---

## [Decision Letter · Decision Letter 2]

30 Dec 2020

Determinants of the intensity of uremic pruritus in patients receiving maintenance hemodialysis: A cross-sectional study

PONE-D-20-18098R2

Dear Dr. Zhao,

We’re pleased to inform you that your manuscript has been judged scientifically suitable for publication and will be formally accepted for publication once it meets all outstanding technical requirements.

Kind regards,

Girish Chandra Bhatt, MD, FASN

Academic Editor

PLOS ONE

Additional Editor Comments (optional):

In the abstract section, please remove numbering as 1), 2) and 3).

Reviewers' comments:

Reviewer's Responses to Questions

**Comments to the Author**

1. If the authors have adequately addressed your comments raised in a previous round of review and you feel that this manuscript is now acceptable for publication, you may indicate that here to bypass the “Comments to the Author” section, enter your conflict of interest statement in the “Confidential to Editor” section, and submit your "Accept" recommendation.

Reviewer #1: All comments have been addressed

2. Is the manuscript technically sound, and do the data support the conclusions?

Reviewer #1: Yes

3. Has the statistical analysis been performed appropriately and rigorously? 

Reviewer #1: I Don't Know

4. Have the authors made all data underlying the findings in their manuscript fully available?

Reviewer #1: Yes

5. Is the manuscript presented in an intelligible fashion and written in standard English?

Reviewer #1: Yes

6. Review Comments to the Author

Reviewer #1: (No Response)

7. PLOS authors have the option to publish the peer review history of their article (what does this mean?). If published, this will include your full peer review and any attached files.

Reviewer #1: No

---

## [Editor Report · Acceptance letter]

11 Jan 2021

PONE-D-20-18098R2 

Determinants of the intensity of uremic pruritus in patients receiving maintenance hemodialysis: A cross-sectional study 

Dear Dr. Zhao:

I'm pleased to inform you that your manuscript has been deemed suitable for publication in PLOS ONE. Congratulations! Your manuscript is now with our production department. 

Kind regards, 

on behalf of

Dr. Girish Chandra Bhatt 

Academic Editor

PLOS ONE